# Experimental Epileptogenesis in a Cell Culture Model of Primary Neurons from Rat Brain: A Temporal Multi-Scale Study

**DOI:** 10.3390/cells10113004

**Published:** 2021-11-03

**Authors:** Janos Jablonski, Lucas Hoffmann, Ingmar Blümcke, Anna Fejtová, Steffen Uebe, Arif B. Ekici, Vadym Gnatkovsky, Katja Kobow

**Affiliations:** 1Department of Neuropathology, Universitätsklinikum Erlangen, Friedrich-Alexander-Universität Erlangen-Nürnberg (FAU), 91054 Erlangen, Germany; janosj@web.de (J.J.); lucas.hoffmann@uk-erlangen.de (L.H.); ingmar.bluemcke@uk-erlangen.de (I.B.); 2Department of Psychiatry and Psychotherapy, Universitätsklinikum Erlangen, Friedrich-Alexander-Universität Erlangen-Nürnberg (FAU), 91054 Erlangen, Germany; Anna.Fejtova@uk-erlangen.de; 3NGS Core Unit, Institute of Human Genetics, Universitätsklinikum Erlangen, Friedrich-Alexander-Universität Erlangen-Nürnberg (FAU), 91054 Erlangen, Germany; Steffen.Uebe@uk-erlangen.de (S.U.); Arif.Ekici@uk-erlangen.de (A.B.E.); 4Department of Epileptology, University Hospital Bonn, 53127 Bonn, Germany; Vadym.Gnatkovsky@ukbonn.de

**Keywords:** epilepsy, in vitro, epigenetic, DNA methylation

## Abstract

Understanding seizure development requires an integrated knowledge of different scales of organization of epileptic networks. We developed a model of “epilepsy-in-a-dish” based on dissociated primary neuronal cells from neonatal rat hippocampus. We demonstrate how a single application of glutamate stimulated neurons to generate spontaneous synchronous spiking activity with further progression into spontaneous seizure-like events after a distinct latency period. By computational analysis, we compared the observed neuronal activity in vitro with intracranial electroencephalography (EEG) data recorded from epilepsy patients and identified strong similarities, including a related sequence of events with defined onset, progression, and termination. Next, a link between the neurophysiological changes with network composition and cellular structure down to molecular changes was established. Temporal development of epileptiform network activity correlated with increased neurite outgrowth and altered branching, increased ratio of glutamatergic over GABAergic synapses, and loss of calbindin-positive interneurons, as well as genome-wide alterations in DNA methylation. Differentially methylated genes were engaged in various cellular activities related to cellular structure, intracellular signaling, and regulation of gene expression. Our data provide evidence that a single short-term excess of glutamate is sufficient to induce a cascade of events covering different scales from molecule- to network-level, all of which jointly contribute to seizure development.

## 1. Introduction

Epilepsy is not a single disease, but highly heterogeneous with diverse clinical syndromes, associated etiologies (i.e., structural, genetic, infectious, metabolic, immune, unknown), and many different mechanisms contributing to the epileptic phenotype and known frequent comorbidities. Temporal lobe epilepsy (TLE) is the most common epilepsy syndrome in adults. Hippocampal sclerosis (HS) is a frequent structural brain lesion identified in patients with drug-resistant focal epilepsy submitted to surgical treatment [1]. It is histopathologically characterized by segmental neuronal cell loss in the hippocampal formation [2]. Clinical disease presentation in patients with TLE often follows a characteristic pattern. An initial precipitating injury is reported in many of these patients, such as prolonged febrile seizures or status epilepticus (SE), followed by a clinically silent latency period before the onset of spontaneous, recurrent, and often drug-resistant seizures [3]. A prevailing hypothesis to best anticipate this phenomenon is secondary epileptogenesis [4,5]. A precipitating injury triggers a molecular cascade turning a normal brain into an epileptic disease condition [6]. A growing number of experimental studies specifically addressed the structural and molecular basis of specific disease pathomechanisms in epileptogenesis, e.g., the imbalance between excitatory and inhibitory neurotransmission [7], the role of fast-spiking interneurons in seizure initiation and network synchronization [8], aberrant neuronal circuitries and plasticity [9], seizure-induced aberrant neurogenesis [10,11], neuroinflammation [12,13], or energy metabolism [14], amongst many other more. Epigenetic alterations were more recently also proposed as a fundamental pathomechanism in epilepsy. Several studies have identified locus-specific and genome-wide alterations in DNA methylation patterns in experimental epilepsy models [15,16] or human surgical brain samples [17,18]. Aberrant DNA methylation was linked to the compromised execution of gene expression programs and thereby suggested to account for many known structural and functional changes in the epileptic brain [15,16].

While seizures are commonly defined as the clinical manifestation of an abnormal, excessive, hypersynchronous discharge of a population of cortical neurons [19], it remains unclear to date what the minimum cellular and molecular requirements are that induce and promote the complex sequence of events leading to seizure development. Many experimental animal models have been developed in the past to induce and study acute seizures or epileptogenesis [20], e.g., following neonatal hyperthermia [21,22], hypoxia [23,24,25,26,27], traumatic brain injury [28,29,30], tetanus toxin injection [31,32,33], pilocarpine induced SE [34,35], kainic acid induced SE [36], or electrical kindling [37]. However, animal models have a limited ability to recapitulate human pathologies even though they appear to symptomatically mimic human disease. Furthermore, they are highly cost and labor- and time-consuming. Additionally, the complexity of the mammalian brain limits the possibilities of addressing questions of cause and consequence, and it remains challenging to attribute specific disease pathomechanisms over time to individual cell populations. In vitro models, including acute hippocampal slice preparations, organotypic hippocampal slice cultures [38], and iPSC-based brain organoid models [39] may be used to address some of these limitations. Moreover, dissociated primary neuronal cultures and neuronal cell lines are used to study basic electrophysiological properties and screen, e.g., for neurotoxicity of certain compounds [40,41,42].

We recently developed a simple and powerful cell culture model of epileptogenesis using a single transient glutamate-induced neuronal mass discharge to transform a primary dissociated cell culture from neonatal rat hippocampus into a repetitively firing synchronous seizure-like network, i.e., “epilepsy-in-a-dish” [43]. To understand seizure development, an integrated understanding of different scales of organization of epileptic networks across time is needed. However, research domains in epilepsy are commonly studied either alone or only at a single time point. The present multi-scale study aimed to quantitatively investigate network and single-cell activity and structural remodeling processes, including dendritic sprouting with or without new synapse formation. We further used whole-genome bisulfite sequencing (WGBS) to study the regulatory role of DNA methylation on biological pathways implicated in the control of cell structure and function. With this model, we can monitor and interfere with the timeline of molecular, structural, and functional changes during epileptogenesis at the neuronal cell and network level.

## 2. Materials and Methods

### 2.1. Animals and Tissue Preparation

Animal and tissue preparation was performed as previously described [43]. In short, adult male and female Wistar rats were obtained from Charles River (Sulzfeld, Germany), bred and maintained at the local animal housing facility in breeding cages under controlled environmental conditions (12 h light/dark cycle, 20–23 °C, 50% relative humidity, drinking and feeding ad libitum). Male and female rat pups (P0–P2) were used. All animal experiments were approved by the local animal care and use committee (TS-1/13) and followed the European Communities Council Directive and German Animal Welfare Act (54–2532.1-23/09, Directive 2010/63/EU).

### 2.2. Cell Culture

Primary neuronal cell cultures from rat neonatal hippocampus were prepared as described before [43]. Briefly, cell suspensions from dissociated newborn rat hippocampi (P0–P2) were preplated for 1 h at 37 °C, 5% CO_2_ onto an uncoated flask to allow settlement of glial cells and their adhesion to the flask. Remaining cells from the supernatant were harvested at 800 rpm for 8 min at room temperature. Cell pellets were resuspended and cultured in serum-free Neurobasal-A medium supplemented with 2% B27, 0.5 mM GlutaMAX, and 1% penicillin–streptomycin (all Life Technologies, Darmstadt, Germany). Cells were plated on poly-D-lysine coated coverslips (Ø 1cm; Greiner Bio-One, Kremsmünster, Austria) at a density of 2.5 × 10^5^ cells/well. Cells were maintained at 37 °C in a fully humidified incubator with 5% CO_2_. After 24 h, 6 µM Cytosine β-D-arabinofuranoside hydrochloride (AraC; Sigma-Aldrich, Taufkirchen, Germany) was added to inhibit proliferation of remaining glial cells. Neurons were continuously maintained in dispersed culture with their original media supplemented with AraC. After 12 days in vitro, the original culture medium was transiently replaced by physiological treatment solution (10 mM HEPES (pH 7.4), 145 mM NaCl, 10 mM glucose, 1 mM MgCl_2_, 2.5 M KCl, 2 mM CaCl_2_, 2 µM glycine) with or without 10 µM glutamate [44,45]. Cultures were then washed three times, resupplied with the original culture medium with AraC, and kept at 37 °C in a fully humidified incubator containing 5% CO_2_ until further experiments.

### 2.3. Viability Assay

Cellular viability was determined 7 and 15 days after glutamate treatment with propidium iodide (PI) and fluorescein diacetate (FDA). Cells were incubated with 500 nM PI and 160 nM FDA for 10 min in Neurobasal A medium at 37 °C. Semiquantitative measurements of PI/FDA-stained cells were performed using a microcomputer imaging system (ColorView II CCD camera, Cell^F imaging software) equipped to an Olympus XI70 microscope. PI- and FDA-positive cell bodies were tagged on the computer screen and manually counted in four independent visual fields at 20× objective magnification. Cell viability was routinely determined in glutamate-treated cultures and time-matched sham controls.

### 2.4. Immunofluorescence

Immunofluorescence staining for Sholl analysis was performed 1, 3, 5, 7, and 15 days after glutamate or sham treatment, for synapse quantification after 3 and 7, as well as for interneuron detection 7 days following initial treatment. Neurons cultured on coverslips were fixed in 4% paraformaldehyde and blocked in tris-buffered saline (TBS; 0.05 M Tris-Cl (pH 7.6)) with 1% bovine serum albumin (BSA; Biochrom, Berlin, Germany), 2% fish skin gelatin (Sigma-Aldrich, Taufkirchen, Germany) and Triton X-100 (0.1%; Sigma-Aldrich) for 2 h at room temperature. Primary antibodies were diluted in blocking solution and applied overnight at 4 °C. Secondary fluorophore-conjugated antibodies were diluted in blocking solution and applied for 4 h at room temperature. Nuclei were dyed with 4′,6-diamidino-2-phenylindole (DAPI; Sigma-Aldrich, Taufkirchen, Germany, 1:1000) diluted in blocking solution. For Sholl analysis, primary chicken anti-Map2 (Abcam, Cambridge, UK, Cat# ab5392, RRID:AB_2138153, 1:2000) and mouse anti-Gfap (Cell Signaling Technology, Frankfurt a.M., Germany, Cat# 3670, RRID:AB_561049, clone GA5, 1:300) and secondary Alexa-647 donkey anti-mouse (Abcam, Cambridge, UK, Cat# ab150107, RRID:AB_2890037, 1:1000) and Alexa-488 goat anti-chicken (Thermo Fisher Scientific, Bremen, Germany, Cat# A-11039, RRID:AB_2534096, 1:1000) were used. For interneuron quantification, primary rabbit anti-Calbindin (Synaptic Systems, Göttingen, Germany, Cat# 214 002, RRID:AB_2068199) and secondary Alexa-555 goat anti-rabbit (Thermo Fisher Scientific, Bremen, Germany, Cat# A-21428, RRID:AB_2535849, 1:1000) were used. For synapse quantification, primary rabbit anti-Vglut1 (Synaptic Systems, Göttingen, Germany, Cat# 135 308, RRID:AB_2864787, 1:500), chicken anti-Vgat (Synaptic Systems, Göttingen, Germany, Cat# 131 006, RRID:AB_2619820, 1:500), and mouse anti-Bassoon (Enzo Life Sciences, Lörrach, Germany, Cat# SAP7F407, RRID:AB_2313990, 1:300), together with secondary Alexa-488 goat anti-chicken and Alexa-555 goat anti-rabbit (Thermo Fisher Scientific, Bremen, Germany, Cat# A-21428, RRID:AB_2535849, 1:1000), as well as Alexa-647 donkey anti-mouse (Abcam, Cambridge, UK, Cat# ab150107, RRID:AB_2890037, 1:1000) were used.

### 2.5. Sholl Analysis

Immunofluorescence microscopy for Sholl analysis was performed with the Zeiss MN Observer microscope (Zeiss, Jena, Germany) and ZEN 2012 SP5 software. Singular neurons were tracked in the periphery of cultures, avoiding overlay of different cells and false-positive attribution of dendrites. All images were calibrated to scale for Sholl analysis afterwards. After processing of images, we used the Sholl analysis plug-in v3.7.0 [46] for Fiji software v1.51s [47] to describe dendritic arbors on singular neurons. A total of 60 neurons were analyzed per time point after glutamate treatment with 30 neurons per treatment, with or without glutamate, respectively.

### 2.6. Synapse Quantification

Synapses were visualized using a Zeiss Observer Z1 LSM 780 microscope (Zeiss, Jena, Germany) equipped with 405/488/555/647 HXP 120 V Fluorescence lamp, 405/488/555/647 LASOS Argon Laser, and 594 Zeiss In Tune Laser, using ZEN 2012 SP5 software. Images were taken with a 1 mm pinhole using a 63× objective covering an area of 134 µm^2^. All images were acquired under standardized conditions. For synapse quantification, five non-overlapping images per coverslip and six independent preparations per experimental condition were used. Co-localized presynaptic (Bassoon-positive) and postsynaptic (Vglut1- or Vgat-positive) puncta within a mask were counted manually using the cell counter plug-in for Fiji (1.51s “Multi-point” tool provided by Fiji). In a second approach, we used the automated Fiji plug-in Synapse Counter. During quantification, settings for synapse counting were uniformly chosen for all images within an experiment. Results of automatic puncta quantification were always validated by visual inspection and manual counting for a subset of images. All coverslips were coded and processed in a blinded manner. We normalized all puncta counts to the median number of puncta detected on control neurons for a particular coverslip to combine data acquired from different coverslips in different experiments. An average of 50,000 synapses was analyzed per experimental condition and time point.

### 2.7. Calcium Imaging

Calcium imaging was performed 1, 3, 5, 7, 9, and 15 days after glutamate or sham treatment. Cultures were incubated for 30 min at 37 °C with 5 µM Fluo-8 AM (Abcam, Cambridge, UK,) in 1 µg/µL DMSO. The osmolarity of modified Cohen’s Recording Solution (1x SBM: 144 mM NaCl, 2.5 mM KCl, 2.5 mM CaCl_2_, 2.5 mM MgCl_2_, 10 mM Glucose, 10 mM HEPES, pH 7.4, 320 mOsm; [48]) was adjusted to the medium’s osmolarity and 5 µM Fluo-8 AM was added. After incubation, coverslips were placed into a perfusion chamber filled with 450 µL recording solution. Nikon Eclipse Ti’s inverted fluorescence microscope (Nikon, Tokyo, Japan) with a Lambda SC shutter was used for the recording. An area of 819.2 µm^2^ was detected with a Plan Apo 10×, 0.45 NA objective, and perfect focus system in every experiment. Fluo-8 AM was excited by Nikon Intensilight C-HGFI Lamp (Neutral Density Filter 16) through a 455–485 nm excitation and a 495 nm dichroic long-pass mirror. Emitted light was passed through a 488 nm filter Cube GFP (Semrock, Rochester, NY, USA). Detection was performed with a cooled EM-CCD camera (iXonEM DU-885; DU-887, Andor Oxford Instruments, Belfast, Northern Ireland, UK). The culture was recorded with the NIS Elements (Nikon, Tokyo, Japan) software in two different settings: no binning and 200 ms exposure time or 2 × 2 binning and 50 ms exposure time. For all experimental conditions, 5–10 biological replicates from at least three different preparations were analyzed.

Cellular calcium signals were quantified as the average light intensity of manually selected regions of interest (ROI; Fiji 1.51s) covering cell bodies. Per-pixel standard deviation was used to identify neurons (ROI) with significant changes in calcium signal. Signals from 80–150 neurons were analyzed in every recording session. Individual traces were exported and analyzed in a custom-developed software designed by V.G. in LabView (National Instruments, Austin, TX, USA) [49].

### 2.8. Whole Genome Bisulfite Sequencing

WGBS paid service was performed at the Core Unit NGS (FAU Erlangen, Germany) and included DNA extraction, bisulfite conversion, library preparation, sequencing, and data analysis. Sequencing was performed with Illumina HiSeq2500 (Illumina, San Diego, CA, USA). Raw paired-end FASTQ files (per sample, flowcell, read direction, and lane) were trimmed with Trim Galore v0.6.2 using cutadapt v2.3, autodetecting and removing Illumina adapter sequences. Trimmed FASTQ files were then merged with zcat/pigz into a single FASTQ file pair per sample. For bisulfite-sensitive alignment and differential methylation calling, Bismark v0.22.1 with hisat2 v2.1.0 mapping was used [50]. Reads were aligned to the rn6 reference genome, with Ensembl rnor6.0.95 as an annotation reference. Differential methylation was examined using the Bioconductor package DSS v2.32.0 [51] from the methylation counts as reported by Bismark. Upon pairwise comparison of biological groups, differentially methylated positions (DMPs) were called and exported (FDR < 0.05). Then, differentially methylated regions (DMRs) with a minimum number of 4 consecutive DMPs were called using the bumphunter package [52]. DMRs (*p* < 0.001) that mapped to enhancers, promotors (2 kb upstream of transcriptional start site/TSS), or gene bodies were used to identify differentially methylated genes (DMGs). The biological function of gene networks was analyzed using Qiagen Ingenuity Pathway Analysis (IPA) software (Qiagen, Hilden, Germany).

### 2.9. Statistics

Statistical analysis was performed using GraphPad Prism7 (GraphPad Software, San Diego, CA, USA). All points shown in this work represent an average from at least three independent measurements in three independent preparations, including epileptogenic and sham control cultures obtained from the same preparations (i.e., sister cultures). Depending on sample size and distribution, paired or unpaired two-tailed t-test (2 groups), ANOVA (≥3 groups), or respective non-parametric tests and corrections for multiple comparisons were performed. Details are provided in the respective Results section. In all figures, asterisks indicate a level of significance of *p* < 0.05, and error bars and specified values represent mean and standard error of the mean (SEM), if not stated otherwise.

## 3. Results

This section may be divided by subheadings. It should provide a concise and precise description of the experimental results, their interpretation, as well as the experimental conclusions that can be drawn.

### 3.1. “Epilepsy-in-a-Dish”

We previously developed an in vitro model of epileptogenesis based on a self-organized network of interconnected neurons and a few glial cells [43]. Starting from dissociated primary neuronal cells from neonatal rat hippocampus, we showed that a single and transient (10 min) application of physiological levels of 10 µM glutamate stimulated neurons to generate spontaneous synchronous spiking activity progressing into spontaneous seizure-like events (SLE). However, a comprehensive characterization and temporal resolution of network activity dynamics following glutamate injury were missing yet. In the present study, we performed a temporal series of live cell calcium imaging to functionally characterize our model and quantify the dynamics of the potential epileptogenic process. Calcium imaging using Fluo8-AM green fluorescent calcium-binding dye was performed 1, 3, 5, 7, 9, and 15 days after glutamate injury (Figure 1), and the neuronal activity of up to 150 cells per visual field was quantified. Neuronal cultures showed no synchronous activity one day after treatment. After three days, single spikes were detected in individual neurons. At five days after glutamate treatment, synchronous spikes, polyspikes, and a few SLEs were observed. At seven days, 60–70% of all glutamate-treated cultures showed SLEs. Almost all cultures showed synchronous spiking activity compared with time-matched sham-treated controls, which remained silent (Figure 1a,b). SLEs became significantly longer in the second week after glutamate treatment, suggesting further progression of the epileptogenic process (Mann–Whitney test, mean SLE duration_1st week_ = 49 ± 10 s, *n* = 18, SLE duration_2nd week_ = 111 ± 6 s, *n* = 11, *p* = 0.0014; Figure 1c).

Computational analysis and visual comparison of in vitro SLEs with human intracranial stereo-EEG recordings of seizures identified strong similarities and a related sequence of events with defined onset, progression, and termination (Figure 1d). SLEs typically showed three prominent features: a build-up phase with increasing spiking activity, followed by a direct current (DC) shift with on-top oscillatory activity starting from high frequency then slowing down (=chirps in power spectrum), and post-SLE depression (Figure 1b,d).

### 3.2. Structure Determines Function

We next thought to analyze whether the functional differences between glutamate-treated and control cultures resulted from structural changes of cultured cells or altered cellular assembly of the networks under study. Therefore, we monitored cultured neurons (Map2 immunoreactivity) for up to 15 days after glutamate treatment and quantified neurite length and sprouting using Sholl analysis (Figure 2). With time, glutamate-treated neurons displayed significantly longer processes (paired two-tailed t-test, mean length_Ctrl1d_ = 241.8 ± 70.2 µm, length_Glu1d_ = 263.1 ± 76.9 µm, *p* = 0.33; length_Ctrl3d_ = 185.9 ± 59.8 µm, length_Glu3d_ = 241.1 ± 58.7 µm, *p* = 0.0003; length_Ctrl5d_ = 187.6 ± 54.9 µm, length_Glu5d_ = 241.1 ± 68.9 µm, *p* = 0.0014; length_Ctrl7d_ = 227.4 ± 37.2 µm, length_Glu7d_ = 303.8 ± 74.0 µm, *p* < 0.0001; length_Ctrl15d_ = 272.9 ± 68.5 µm, length_Glu15d_ = 289.5 ± 101.7 µm, *p* = 0.62).

When we plotted the number of neurite intersections against distance from soma: neurons following glutamate treatment displayed fewer intersections close to the soma, but more intersections with increasing distance from the soma, as compared with time-matched sham-treated controls (paired two-tailed *t*-test, mean branching_Ctrl1d_ = 7.0 ± 4.3, _Glu1d_ = 4.9 ± 3.3, *p* < 0.0001; _Ctrl3d_ = 4.6 ± 2.8, _Glu3d_ = 3.5 ± 2.3, *p* < 0.0001; _Ctrl5d_ = 4.8 ± 2.9, _Glu5d_ = 5.2 ± 3.6, *p* = 0.62; _Ctrl7d_ = 7.8 ± 4.7, _Glu7d_ = 5.9 ± 3.8, *p* < 0.0001; _Ctrl15d_ = 5.9 ± 4.1, _Glu15d_ = 4.0 ± 3.0, *p* < 0.0001; Figure 2a,b).

We next performed in-depth manual and automated quantification of glutamatergic and GABAergic synapses. We identified a relative loss of active GABAergic synapses stained for vesicular GABA transporter (Vgat) over excitatory synapses stained positive for vesicular glutamate transporter 1 (Vglut1) seven days after glutamate treatment compared with time-matched sham controls (unpaired two-tailed t-test, %Vgat_Ctrl7d_ = 43.4 ± 3.2, %Vgat_Glu7d_ = 24.0 ± 3.1, *p* < 0.001). This difference was not yet detectable three days after epileptogenesis induction, i.e., following glutamate exposure (unpaired two-tailed t-test, mean %Vgat_Ctrl3d_ = 35.0 ± 1.2, %Vgat_Glu3d_ = 37.9 ± 3.5, *p* > 0.05). Only synapses co-labeled for Bassoon (Bsn, red), a scaffolding protein involved in organizing the presynaptic cytoskeleton of active axon terminals, and Vgat or Vglut1 were included in the measurements (Figure 3a–c, Appendix A).

To examine whether the loss in GABAergic synapses resulted from a loss in GABAergic interneurons or rather from a change in synaptic plasticity, immunofluorescence staining was performed for neuronal microtubule-associated protein 2 (Map2), glial fibrillary acidic protein (Gfap), and the interneuronal marker protein Calbindin (Calb1). A general decline in Calb1-positive cells relative to overall cell numbers was observed in cultures seven days after glutamate treatment (unpaired two-tailed t-test, %Calb1_Ctrl7d_ = 43.1 ± 1.6, %Calb1_Glu7d_ = 36.7 ± 2.5, *p* = 0.031; Figure 3d,e). Taken together, the data suggest a time-dependent shift towards increased excitability.

### 3.3. Epigenetic Regulation of Causative Gene Networks

Activity-driven alterations of gene expression patterns induce structural and functional changes in neurons, critical for long-term adaptations of brain circuits and indispensable for proper brain function [53,54]. To test whether functional and structural changes in our model could be linked with potentially gene-regulatory DNA methylation signatures, we performed whole-genome bisulfite sequencing (WGBS) in primary neuronal cell cultures seven days after glutamate treatment (n = 4) and in time-matched controls (n = 5).

From pairwise comparisons, 538 differentially methylated positions (DMP) were identified with an FDR < 0.05 (Figure 4a). Principal component analysis for dimensionality reduction and hierarchical clustering showed that all samples clustered in their respective treatment groups (control—blue, glutamate–red; Figure 4b). Both increase and decrease in DNA methylation across the genome were observed in the disease model (Figure 4b,c). Next, differentially methylated regions (DMRs, *p* < 0.001, n = 279) were called, which uniquely mapped to 130 differentially methylated genes (DMGs). We then performed functional pathway analysis using Qiagen’s ingenuity pathway analysis software. There was an enrichment of DMG in the axonal guidance signaling pathway in line with the dominant structural changes that we detected in neuronal networks following glutamate treatment. Next, functional interaction networks of proteins encoded by hyper- (green) and hypomethylated genes (red) were identified and mapped to the cell as a structural and functional unit. Thereby the spatial location of protein nodes was shifted into their corresponding subcellular locations (Figure 4c). DMGs and their encoded proteins were found to be part of a broad range of cellular structures and functional processes including extracellular matrix (ECM) structure; plasma membrane composition, including membrane-bound and soluble signaling factors, ion channels, transporters, and receptors; components of the actin cytoskeleton; intracellular signaling and metabolism; or proteins involved in chromatin structure and function (Table 1). Many of these proteins play a significant role in nervous system development, function, and disease (Figure 4c, pink bordered icons; Table 1). Our data suggest that genomic DNA methylation is altered early on in the epileptogenic process and may play a central role in regulating pro-epileptogenic neuronal gene expression and downstream cellular structure and function throughout the entire disease process.

## 4. Discussion

Epileptogenesis, i.e., the process that leads to the development of spontaneous recurrent seizures and disease progression, cannot be studied in the human brain in situ. Only in a subset of patients undergoing epilepsy surgery for treatment of their focal drug-resistant epilepsy do we have access to human brain tissue. However, the described molecular, structural, and functional changes in such tissue are representative of a late (chronic) disease stage and do not allow conclusions to be drawn on when these changes occurred. Therefore, experimental models are needed to study the disease course. Here, we describe a simple but powerful model providing insights into the full process of epileptogenesis in an exceptionally short time frame covering all steps from initial precipitating injury to silent latency period, development of spontaneous recurrent seizure-like events (SLEs), and disease progression. We performed an integrated study addressing multiple scales from molecules to cellular and network structure and neurophysiological function. Our data show striking and broad similarities between our cellular model system with human epilepsy across scales including similar seizure pattern and duration, cell structure, and a specific phenotype-associated gene-regulatory epigenetic profile.

Rodent models of chronic epilepsy have been used extensively in the past to study epileptogenesis. Still, they have been proven to be labor- and cost-intensive and challenging to study in sufficient temporal, cellular, and molecular resolution. Acute or organotypic slice cultures are often used for more detailed studies on seizure mechanisms, as they sustain much of the necessary circuitry. While acute preparations are viable for a few hours and valuable for studying acute provoked seizures, organotypic slice cultures are sustainable for multiple weeks. They can be used to study temporal processes, including circuit development, synaptic plasticity, and axonal sprouting. However, interpretation of molecular data from brain tissue with its high cellular heterogeneity remains challenging, as results cannot easily be attributed to a specific cell type. Approaches that help address this issue include cell sorting before molecular analysis, microfluidics and subsequent single-cell sequencing, or culturing select cell populations, all of which have specific technical and biological advances and limitations. Here, we used primary neuronal cells from rat HC that, after maturation in vitro, self-assembled into 3D functional networks (without the support of an artificial 3D matrix) and developed spontaneous recurrent epileptiform activity following a single transient stimulus with physiological levels of glutamate. With a stable 92–96% neuronal phenotype, our cultures could be used for a nearly unbiased study of neuronal features of seizure generation.

Epileptogenesis was monitored following glutamate exposure using live-cell calcium imaging at different time points. After 24 h, no specific neuronal activity was detected except for episodic random unsynchronized spikes. At 3 days after the transient glutamate exposure, synchronous spiking activity was detected and remained prominent at 5, 7, 9, and 15 days. SLEs emerged 5 days after treatment and were maintained in the 7-, 9-, and 15-day cultures. Cell type attribution of detected signals was always confirmed by Map2/Gfap immunofluorescence of all calcium-imaged cultures. We identified specific but extremely rare events of glial calcium activity (i.e., low frequency, high amplitude, slowly propagating along glial processes), which only affected individual cells and never occurred in temporal relation with synchronized spiking or SLE development across the network. SLEs in our cultures showed strong similarities with human intracranial EEG seizures, both in terms of duration, stability for around 1–2 min, and according to seizure pattern. The similarities between experimental and human seizure power spectra were striking despite the differences in resolution. Our results indicate the development of spontaneous recurrent epileptiform activity after a short silent period and progressive worsening with more prolonged SLEs over time, thus recapitulating the presumed general pathogenic time course of epileptogenesis. In our study design, no continuous measurement of epileptiform activity was possible. Preparations of cultures on (high-density) multi-electrode arrays would allow measurements of field potentials to examine network activity at higher temporal resolution, however not at single-cell resolution. Other complementary studies could involve whole-cell patch-clamp approaches, enabling manipulations of individual cells and determining downstream small network effects if several cells over the network are monitored simultaneously.

Structural changes of neuronal cells during epileptogenesis were studied using Map2 immunofluorescence staining and Sholl analysis. The neurite length of neurons in glutamate-treated cultures was significantly increased early on in the epileptogenic process. This notable change in cellular structure persisted over time, together with a higher degree of terminal branching. Altered cell morphology in our model matched structural changes previously described ex vivo in humans [55] and rats [16], where dendritic and axonal sprouting have been identified and suggested to contribute to the proepileptogenic reorganization of hippocampal circuits. In addition, we identified an increase in active glutamatergic over GABAergic synapses at 7 but not 3 days after glutamate treatment compared with time-matched sham controls. Variability in relative Vglut1- and Vgat-positive synapse quantities may have been related to individual preparations. We aimed to control this technical factor by analyzing sister cultures when comparing glutamate-treated to sham controls. The observed shift in synaptic composition towards an increase in active glutamatergic synapses was evident, however, and may contribute to an increase in excitability of cell cultures.

To further investigate DNA methylation changes, a more permanent regulator for gene expression, we performed WGBS of cultures 7 days after initial precipitating injury. Principal component analysis and hierarchical clustering revealed a highly distinct pattern in genomic DNA methylation for glutamate- versus sham-treated neurons. Thereby, our data support extensive epigenomic changes even as short as 7 days after the treatment, confirming previous reports of DNA methylation dynamics in epileptogenesis [15,16,18,43,56,57]. Our data further support the notion of a leading regulatory role in pathologic gene expression in epilepsy. Both acquisition and loss of DNA methylation were observed to target genic and intergenic parts of the genome. These findings were in line with previous reports from Kobow et al. in vivo [15,16]. Next, differentially methylated CpGs were clustered to identify biologically meaningful differentially methylated regions (DMR) and then mapped to gene bodies, promoters, and enhancers. Thereby, 130 genes were targeted by differential methylation in epileptogenesis and encoded proteins with a broad range of functions throughout the entire cell, from ECM and cell membrane components to signaling molecules in the cytoplasm and structural proteins of the cytoskeleton, as well as nuclear factors involved in chromatin structure and function (Table 1). Functional pathway analysis specifically highlighted axon and neurite growth. This was in line with the broad structural changes and increased neurite outgrowth that we detected in our model. While we identified no further enrichment of differentially methylated genes (DMGs) in Kyoto Encyclopedia of Genes and Genomes (KEGG) or Reactome pathways, which is likely due to an annotation bias in these databases, ingenuity pathway analysis revealed several functional protein interaction networks. In-depth analysis of specific candidate genes highlighted various roles in brain development, function, and disease, including neurodegenerative disease, autism spectrum disorders, intellectual disability, developmental brain malformations, and epilepsy. We detected hypermethylation of Potassium Voltage-Gated Channel Subfamily H Member 5 (*Kcnh5*), an outward-rectifying, non-inactivating potassium channel that regulates neurotransmitter and hormone release. Diseases associated with *KCNH5* mutations include early infantile epileptic encephalopathy (EIEE). We also found hypermethylation of collagen type XIV, alpha 1 (*Col14a1*), an ECM component that is highly expressed in the brain and has been linked to familial adult myoclonic epilepsy (FAME). We further identified hypermethylation of doublecortin (*Dcx*). Dcx is a microtubule-associated protein involved in neuronal migration. Mutations in human *DCX* have been identified as being causative in double cortex syndrome and other cortical malformations associated with epilepsy. Hence, although we lack a transcriptomic profile of our cultures, these data support a gene regulatory role of DNA methylation, which could explain many of the structural and functional changes seen in our model of epileptogenesis. The diversity of gene functions potentially affected by DNA methylation changes in our model are in line with the multifactorial nature of epilepsy and associated comorbidities and its link to brain development, function, and aging.

Characteristic genomic methylation signatures for different disease stages have been reported in various species, etiologies, and pathologies [15,16,17,18,57,58]. A comprehensive integration and comparison of genomic DNA methylation data obtained in the present study (WGBS), with published data from different rat epilepsy models (Methyl-Seq), and human focal epilepsy (Methyl-Seq, Illumina arrays) has not yet been performed. Such integrated analysis with the aim of identifying common epileptogenic mechanisms would be challenging with regard to species differences, annotation biases, platform variability, normalization algorithms, cutoff points for selecting regulated genes, analysis of different brain structures and cells, use of variable insults to trigger epileptogenesis, selection of time points for sampling after the insult, and characterization of epilepsy phenotype at the time of sampling. However, comparison of DNA methylation profiles from Methyl-Seq in the epileptic hippocampus obtained from three different rat epilepsy models (i.e., pilocarpine, amygdala stimulation, and traumatic brain injury) distinguished controls from epileptic animals but failed to identify a joint methylation signature in all three different models and few regions common to any two models despite the fact that the study design involved control for the same species, same time point/age of sampling, same tissue, same profiling platform, and the same sequencing facility. Debski et al. concluded that their data provided evidence that genome-wide alteration of DNA methylation signatures is a general pathomechanism associated with epileptogenesis and epilepsy in experimental animal models, but the broad pathophysiological differences between models were reflected in distinct etiology-dependent DNA methylation patterns [15]. Monitoring DNA methylation at 7 days after glutamate injury in our model showed that epigenetic alterations were not confined to single genes, extending previous reports of locus-specific epigenetic gene regulation in this model [43]. Instead, our data further confirmed that genome-wide aberrant DNA methylation is a conserved fundamental gene regulatory mechanism in the pathogenesis of epilepsy. Future studies will be needed to identify specific and reproducible DNA-methylation-based biomarkers of epileptogenesis and the risk of developing seizures after an initial precipitating injury—both in vitro and in vivo.

## Figures and Tables

**Figure 1 cells-10-03004-f001:**
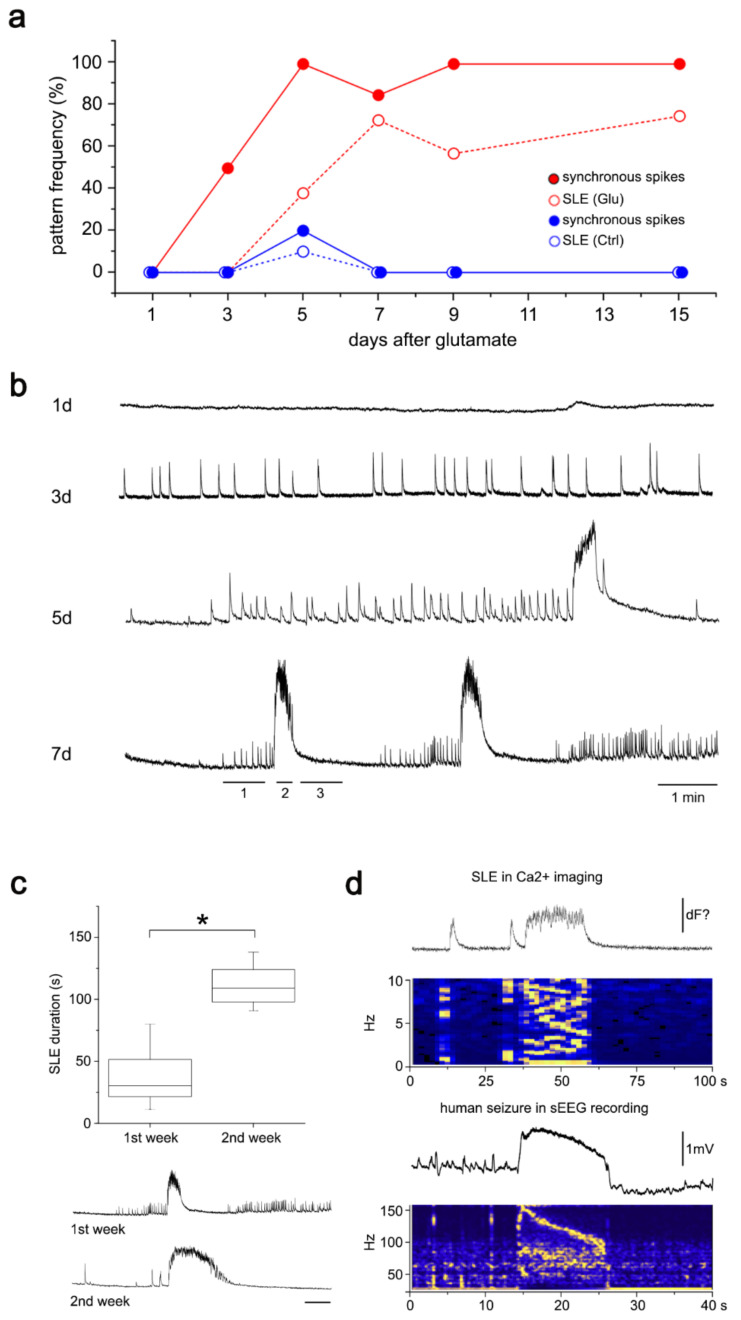
Calcium (Ca^2+^) imaging identified seizure-like activity in vitro. Glutamate-treated rat primary neuronal cultures showed temporal epileptogenic network activity starting after a short latency period with synchronized spiking and later development of seizure-like events (SLE; (**a**,**b**)), which progressed and became longer, showing a duration up to two minutes (**c**). SLEs were characterized by a sharp-onset/sharp-offset transient superimposed on low-voltage fast activity, high-amplitude rhythmic bursts, and postictal depression, thereby showing strong similarities with human stereo-EEG (sEEG) seizure. Traces from individual cells and sEEG contacts are displayed for comparison. Relative power spectrograms summarizing network activity in vitro and in vivo show typical down-chirping (i.e., linear decrease in frequency) (**d**). Ctrl—control (blue), Glu—glutamate (red), sEEG—intracranial stereo-electroencephalogram recording, SLE—seizure-like event. Asterisk (*) indicates significance.

**Figure 2 cells-10-03004-f002:**
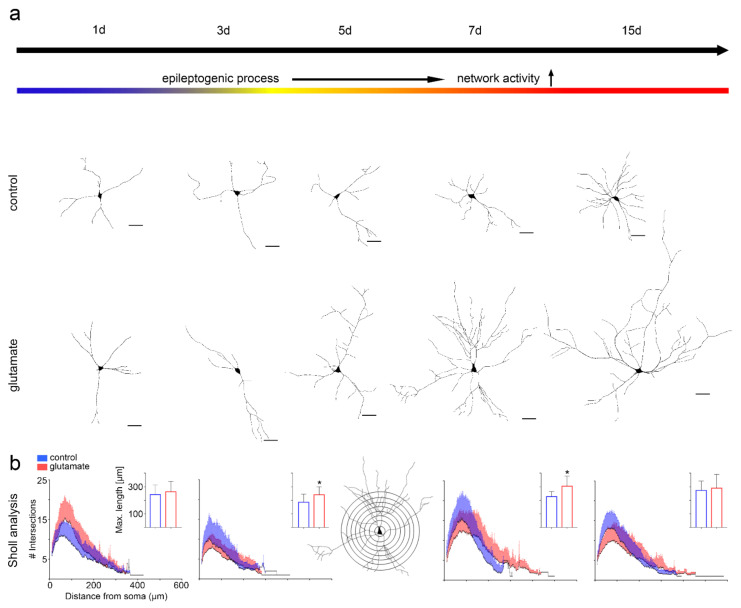
Neuronal morphology in epileptogenesis in vitro. (**a**) Representative images of individual neurons at different time points and sham or glutamate treatment. (**b**) Sholl analysis of 30 captured neurons per time point and treatment. The number of neurite branches or intersections was plotted against the distance from the soma. Schematic of Sholl analysis shown in the middle. Asterisks (*) indicate significance. d—day.

**Figure 3 cells-10-03004-f003:**
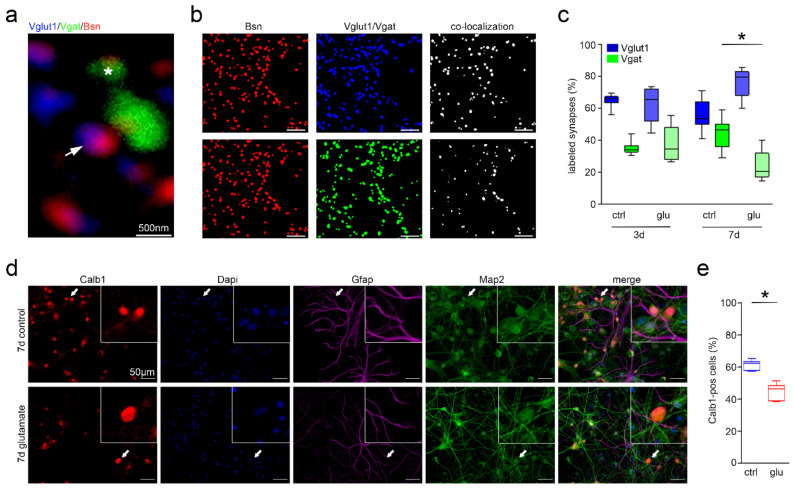
Relative loss of GABAergic synapses and Calbindin-immunoreactive cells. (**a**) Glutamatergic and GABAergic synapses were stained with Vglut1 (blue, white arrow) and Vgat (green, asterisk), respectively, for manual and automated quantification. Only synapses showing co-localization with Bsn (red) were counted. Confocal microscopy at 63×. (**b**) Processed immunofluorescence images for pre- and post-synapses of Fiji plug-in “Synapse Counter” and resulting co-localizations. Scale bars are 500 nm. (**c**) Relative quantification of GABAergic over glutamatergic synapses at 3 and 7 days after glutamate treatment as compared with time-matched sham-treated controls. (**d**,**e**) Relative amount of Calb1-positive neurons compared with Calb1-negative neurons in glutamate and sham-treated cultures after 7 days. Asterisks (*) indicate significance. ctrl—control, d—day, glu—glutamate.

**Figure 4 cells-10-03004-f004:**
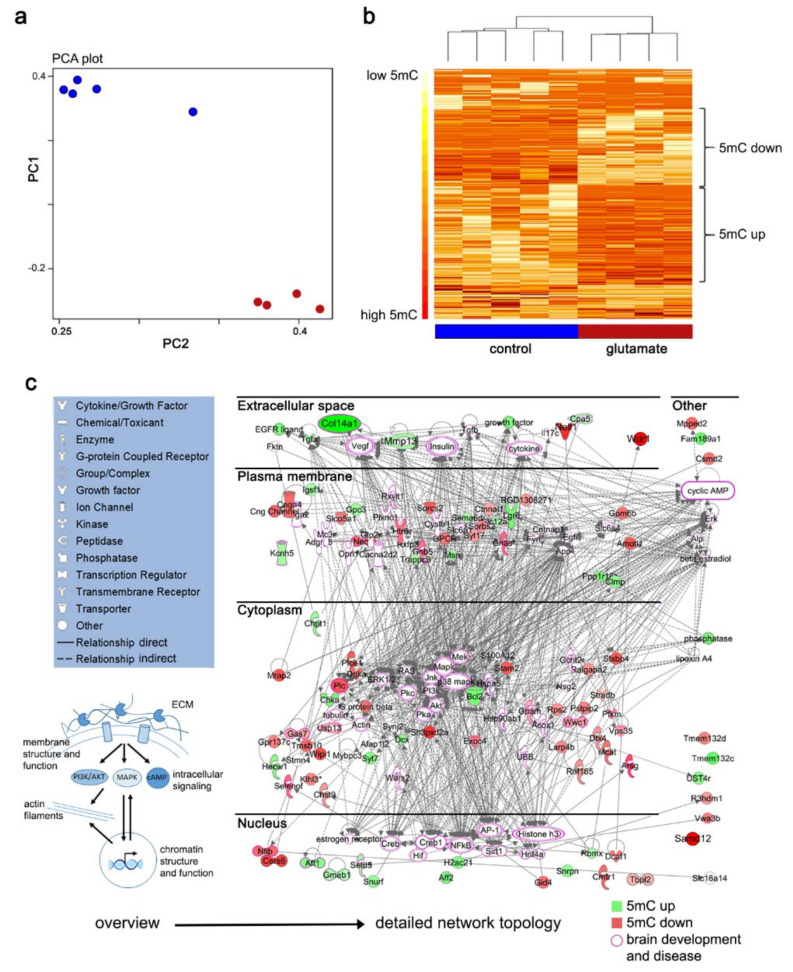
(**a**,**b**) PCA plot and heatmap showing distinct DNA methylation patterns in control (blue) and epileptogenic networks (red). Clustered heatmap of methylation percentages (ranging from 0% to 100 %) of all differentially methylated positions with an F-value of 0.05 or lower. Colors are relative, with darker colors representing higher methylation percentages. (**c**) Functional relationships between differentially methylated genes and their encoded proteins (5mC up = green, 5mC down red), whereby the spatial location of nodes was shifted into their corresponding subcellular locations. Proteins encoded by genes targeted by differential DNA methylation in our analysis mapped to the extracellular space, plasma membrane, cytosol, nucleus, and other. Direct and indirect interactions between DMGs and their respective encoding proteins are shown, illustrating a highly complex interaction network contributing to the epileptic phenotype in glutamate-treated cultures. A simplified overview for orientation is displayed in blue (lower left corner). 5mC—5-methyl-Cytosine, PI3K/AKT—Phosphatidylinositol—4,5—Bisphosphate 3—Kinase / AKT Serine/Threonine Kinase 1, ECM—extracellular matrix, MAPK—Mitogen-Activated Protein Kinase 1, PCA—principal component analysis.

**Table 1 cells-10-03004-t001:** DMGs with a role in brain development, function, and disease. 5mC—5-methyl-Cytosin/ DNA methylation, ADHD—attention deficit hyperactivity disorder, BDNF—brain derived neurotrophic factor, DMG—differentially methylated gene, ECM—extracellular matrix, EIEE—early infantile epileptic encephalopathy, ERK/MAPK—extracellular signal-regulated kinases/ mitogen—activated protein kinase, FAME—familial adult myoclonic epilepsy, GPCR—G—protein coupled receptor, IDD—intellectual developmental disorder, MCD—malformation of cortical development, mTOR—mammalian target of rapamycin.

Gene Name	Symbol	5mC	Compartment	Function	Disease Association
Collagen XIV a 1	*Col14a1*	↑	Extracellular space	ECM component	Epilepsy (FAME)
Metalloproteinase 13	*Mmp13*	↑		ECM component, involved in neuroprotection and neurorepair	Alzheimer’s disease, stroke
Neural EGFL Like 1	*Nell1*	↓		Growth factor, control of cell growth and differentiation, nervous system development	Neuroblastoma, craniosynostosis
Transforming Growth Factor Alpha	*Tgfa*	↑		Growth factor, brain development	Hypothalamic hamartoma
Angiomotin Like 1	*Amotl1*	↓	Plasma membrane	Controls paracellular permeability and maintains cell polarity, tight junctions	ADHD
CUB and Sushi Multiple Domains 2	*Csmd2*	↓		Synaptic transmembrane protein required for neuronal maturation, regulates the development, and maintenance of dendrites and synapses	Psychiatric disease
Catenin Alpha Like 1	*Ctnnal1*	↓		Cadherin binding, scaffolding protein	Hirschsprung disease
G Protein Subunit Beta 5	*Gnb5*	↓		Neuronal signaling	IDD, ADHD
Neuronal Membrane Glycoprotein M6B	*Gpm6b*	↓		Neuronal glycoprotein, involved in neuronal differentiation, myelination, maintenance of actin cytoskeleton, role in neuroplasticity	Depression
5-Hydroxytryptamine Receptor 6	*Htr6*	↓		GPCR, regulates cholinergic neuronal transmission in the brain, binds antidepressants	Alzheimer’s disease, schizophrenia
Potassium Voltage-Gated Channel Subfamily H Member 5	*Kcnh5*	↓		Outward-rectifying, non-inactivating channel regulating neurotransmitter and hormone release	Epilepsy (EIEE, Otahara Syndrome)
Semaphorin 6d	*Sema6d*	↑		Axon pathfinding	
Sorbin and SH3 Domain Containing 2	*Sorbs2*	↓		Part of the actin cytoskeleton, role in dendritic development, memory, stiffness sensing, and contractile force generation	Intellectual disability
Sortilin Related VPS10 Domain Containing Receptor 2	*Sorcs2*	↓		Neuropeptide receptor, binds precursor forms of NGF (proNGF) and BDNF (proBDNF), regulation of dendritic spine density, required for normal neurite branching and extension in response to BDNF, mediates BDNF-dependent synaptic plasticity, long-term depression, and long-term potentiation	Huntington’s disease
Trafficking Protein Particle Complex 9	*Trappc9*	↑		Activator of NF-kappa-B, role in neuronal differentiation	Intellectual disability, MCD
WD Repeat Domain 1	*Wdr1*	↓		Cofilin cofactor, involved in actin cytoskeletal dynamics, activator of NF-kappa-B, role in neuronal differentiation	Glioblastoma, intellectual disability, MCD
BCL2 Apoptosis Regulator	*Bcl2*	↑	Cytoplasm	Suppresses apoptosis in neural cells	Glioma
Doublecortin	*Dcx*	↑		Microtubule organization, neuronal migration, brain development	MCD, epilepsy
Growth Arrest Specific 7	*Gas7*	↓		Expressed primarily in terminally differentiated brain cells, role in neuronal development	
G-protein Coupled Receptor 137c	*Gpr137c*	↓		Regulates Rag and mTORC1 localization and activity	
Synaptotagmin 7	*Syt7*	↑		Ca2+ sensor involved in exocytosis of secretory and synaptic vesicles; short-term synaptic potentiation	Bipolar disorder
Synaptotagmin 17	*Syt17*	↓		Role in regulating fusion of intracellular vesicles with the plasma membrane, controls neurite outgrowth and synaptic plasticity	
Thymosin Beta 10	*Tmsb10*	↓		Important role in the organization of the cytoskeleton, binds to and sequesters actin monomers and therefore inhibits actin polymerization	
WD Repeat Domain, Phosphoinositide Interacting 1	*Wipi1*	↓		Scaffolding protein	Neurodegeneration
WW And C2 Domain Containing 1	*Wwc1*	↓		Regulator of Hippo signaling, regulates collagen-stimulated activation of the ERK/MAPK cascade, plays a role in cognition and memory performance	Intellectual disability
AF4/FMR2 Family Member 1	*Aff1*	↑	Nucleus	Transcription factor	Fragile X intellectual disability, ataxia
AF4/FMR2 Family Member 2	*Aff2*	↑		Transcription factor	Fragile X intellectual disability
ID Complex Subunit 4 Homolog	*Gid4*	↓		Mediator complex component, is required for activation of RNA Pol II transcription by DNA-bound transcription factors	Speech delay, intellectual disability, MCD
H2A Clustered Histone 21	*H2ac21*	↑		Chromatin organization, core component of nucleosome	
Nuclear Factor I B	*Nfib*	↓		Transcription factor, essential for proper brain development	Macrocephaly and mental retardation
RNA Binding Motif Protein X-Linked	*Rbmx*	↑		Regulates pre-mRNA alternative splice site selection, e.g., for Tau protein	
SET Domain Containing 5	*Setd5*	↑		Histone-Lysine Methyltransferase, mediates H3K36me3	Autosomal dominant intellectual disability
Small Nuclear Ribonucleoprotein Polypeptide N	*Snrpn*	↑		pre-mRNA processing; may contribute to tissue-specific alternative splicing	Prader–Willi Syndrome
SNRPN Upstream Reading Frame	*Snurf*	↑			Prader–Willi Syndrome, Angelman Syndrome
Sterile Alpha Motif Domain Containing 12	*Samd12*	↓		DNA-binding molecule of unclear function	FAME
TATA-Box Binding Protein Like 2	*Tbpl2*	↓		Transcription factor	Rolandic epilepsy
R3H Domain Containing 1CUB and Sushi Multiple Domains 2	*R3hdm1Csmd2*	↓		Required for neuronal maturation, regulates development and maintenance of dendrites and synapses	Mild intellectual disability, psychiatric disease
Willebrand Factor A Domain-Containing Protein 3BR3H Domain Containing 1	*Vwa3bR3hdm1*	↓	Other	Role in brain development, involved in apoptotic signaling in neuronal cells. Role in neuron dendritic growth and branching	Spinocerebellar ataxia with intellectual disability, mild intellectual disability
Willebrand Factor A Domain-Containing Protein 3B	*Vwa3b*	↓		Role in brain development, involved in apoptotic signaling in neuronal cells	Spinocerebellar ataxia with intellectual disability

## Data Availability

The WGBS data presented in this study will be made openly available in GEO under accession number GSEXXX.

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
