# Peer review of "Experimental Epileptogenesis in a Cell Culture Model of Primary Neurons from Rat Brain: A Temporal Multi-Scale Study"

_cells, 2021, doi:10.3390/cells10113004_

Round 1

Reviewer 1 Report

The research article entitled "Experimental Epileptogenesis in a Cell Culture Model of Primary Neurons from Rat Brain: A Temporal Multi-Scale Study" reports an in-vitro neuronal culture study wherein ten micromolar glutamate incubation induces seizure-like activity and subsequent structural plasticity that is proposed to be the result of neuronal transcriptional changes. Overall, the article is well written and highly comprehensible. However, I have the following concerns that, if addressed, will help improve this article:

  1. I struggled a bit with the novelty of this research article. Most of the conclusions are well known from the epilepsy literature. Please be specific in how this study helps fill a gap in our knowledge. Is it simply a characterization of an in-vitro model of epilepsy?
  2. Please provide more (extensive) details about the neuron culture methodology such that the study reported here can be independently reproducible. I tried to refer to the publication cited (16) to get more details. However, I could not confirm certain things – Seeding density of neurons and glia and co-culturing protocol, the concentration of AraC introduced, media replacement frequency how much, and composition of replacement media, etc. Was AraC continuously being present during the glutamate addition and later time point? These essential aspects can regulate neurons:glia cell ratio, dendritic complexity, baseline neuronal activity, etc.
  3. Immunofluorescence method: anti-bassoon antibody source/concentration missing.
  4. WGBS sequencing: Results in 3.3 implies “WGBS of neuronal cultures” that could be misleading as there could be glia too. Please edit the language so that its explicit that genome sequencing was performed on neuron-glia and was not specific to either population.
  5. Loss of calbindin with glutamate-treatment: Stats are missing. Please add them to make this conclusion strong.
  6. Discussion, page 16: “With a stable 92-96% neuronal phenotype, our cultures could be used for a nearly unbiased study of neuronal features of seizure generation”. Astrocytes heavily regulate neuronal synaptic communications. Furthermore, astrocytes are essential in the process of epileptogenesis. With >90% neuronal culture, will the neuronal physiology be reliably recapitulated?

Author Response

Reviewer 1

The research article entitled "Experimental Epileptogenesis in a Cell Culture Model of Primary Neurons from Rat Brain: A Temporal Multi-Scale Study" reports an in-vitro neuronal culture study wherein ten micromolar glutamate incubation induces seizure-like activity and subsequent structural plasticity that is proposed to be the result of neuronal transcriptional changes. Overall, the article is well written and highly comprehensible. However, I have the following concerns that, if addressed, will help improve this article:

  1. I struggled a bit with the novelty of this research article. Most of the conclusions are well known from the epilepsy literature. Please be specific in how this study helps fill a gap in our knowledge. Is it simply a characterization of an in-vitro model of epilepsy?

Response: We kindly thank the reviewer for this comment and take the opportunity to further specify and discuss the novelty of this research article. “Epileptogenesis, i.e., the process that leads to the development of spontaneous recurrent seizures and disease progression, cannot be studied in the human brain in situ. Only in a subset of patients undergoing epilepsy surgery for treatment of their focal drug-resistant epilepsy we have access to human brain tissue. However, the described molecular, structural, and functional changes in such tissue are representative of a late (chronic) disease stage and do not allow to draw conclusions on when these changes occurred. Therefore, experimental models are needed to study the disease course” (p.15, ll.392-398). A wide variety of animal models of acquired epilepsy exist, but they are cost, labor, and time consuming limiting their applicability for temporal in depth studies across different scales of organization of epileptic networks. Given that “the best model of a cat is a cat and most likely the same cat” it is crucial to rigorously test, what we actually model to be sure how translatable results are. “Here, we present a simple but powerful model providing insights into the full process of epileptogenesis in an exceptionally short time frame covering all steps from initial precipitating injury to silent latency period, development of spontaneous recurrent SLEs, and disease progression. We performed an integrated study addressing multiple scales from molecules to cellular and network structure, and neurophysiological function. Our data show striking and broad similarities between our cellular model system with human epilepsy across scales including similar seizure pattern and duration, cell structure, and a specific phenotype-associated gene-regulatory epigenetic profile” (p15., ll. 398-406) . It is further the first report of genomic DNA methylation changes in an in vitro model of epileptogenesis and we identified these changes within a week after glutamate treatment, which matches with the temporal evolution of epileptogenicity, but is at the same time remarkable with regards to the dynamics of this epigenetic mark as DNA methylation is often discussed as rather slowly changing and biologically stable mark.

We have adopted the introduction and discussion section of our manuscript to incorporate these thoughts. See also responses below.

  1. Please provide more (extensive) details about the neuron culture methodology such that the study reported here can be independently reproducible. I tried to refer to the publication cited (16) to get more details. However, I could not confirm certain things – Seeding density of neurons and glia and co-culturing protocol, the concentration of AraC introduced, media replacement frequency how much, and composition of replacement media, etc. Was AraC continuously being present during the glutamate addition and later time point? These essential aspects can regulate neurons:glia cell ratio, dendritic complexity, baseline neuronal activity, etc.

Response: We appreciate this critical comment and are happy to clarify. AraC was added to the culture media within 24h of cell preparation and then continuously present in the media. Except for the transient 10 minute treatment with glutamate or sham treatment solution at 12 DIV, culture media were not replaced. Treatment solutions did not contain AraC. The seeding density of neurons and glia was not separately determined. The overall seeding density was 250K cells/well. Cellular composition of cultures was determined two weeks after sham or glutamate treatment, as previously reported identifying a neuronal phenotype in 92-96% of all cells (Kiese et al., 2017). We have now specified our materials and methods section accordingly:

Briefly, cell suspensions from dissociated newborn rat hippocampi (P0-P2) were preplated for 1 h at 37°C, 5% CO2 onto an uncoated flask to allow settlement of glial cells and their adhesion to the flask. The remaining cells from the supernatant were harvested at 800 rpm for 8 min at room temperature. Cell pellets were resuspended and cultured in serum-free Neurobasal-A medium supplemented with 2% B27, 0.5 mM GlutaMAX and 1% penicillin-streptomycin (all Life Technologies, Darmstadt, Germany). Cells were plated on poly-D-lysine coated coverslips (Æ 1cm; Greiner Bio-One, Kremsmünster, Austria) at a density of 2.5x105 cells/well. Cells were maintained at 37°C in a fully humidified incubator with 5% CO2. After 24 h, 6 µM Cytosine β-D-arabinofuranoside hydrochloride (AraC; Sigma-Aldrich) was added to inhibit the proliferation of remaining glial cells. Neurons were continuously maintained in dispersed culture with their original media supplemented with AraC. After 12 days in vitro, the original culture medium was transiently replaced by physiological treatment solution (10 mM HEPES [pH 7.4], 145 mM NaCl, 10 mM glucose, 1 mM MgCl2, 2.5 M KCl, 2 mM CaCl2, 2 µM glycine) with or without 10 µM glutamate [1, 31]. Cultures were then washed three times, resupplied with the original culture medium with AraC, and kept at 37°C in a fully humidified incubator containing 5% CO2 until further experiments.” (p.3, ll. 106-122)

  1. Immunofluorescence method: anti-bassoon antibody source/concentration missing.

Response: As stated in the materials and methods section, the mouse anti-bassoon antibody was purchased from Enzo Life Sciences and used at a dilution of 1:300 (p.3, l.153).

  1. WGBS sequencing: Results in 3.3 implies “WGBS of neuronal cultures” that could be misleading as there could be glia too. Please edit the language so that its explicit that genome sequencing was performed on neuron-glia and was not specific to either population.

Response: Given that there is roughly a 50/50 ratio of neuronal and glial cells over the entire brain (Barthel, Bahney  Herculano-Houzel 2016), which may vary depending on species and brain region under study, we consider a 92-96% neuronal culture sufficiently pure to attribute any molecular changes detected in the present study mainly to neurons. However, to avoid confusion, we changed the wording as follows: “[…] we performed whole-genome bisulfite sequencing (WGBS) in primary neuronal cell cultures seven days after glutamate treatment (n=4) and time-matched controls (n=5).” (p.10, ll. 342-343)

  1. Loss of calbindin with glutamate-treatment: Stats are missing. Please add them to make this conclusion strong.

Response: We now added statistics to the text: “A general decline in Calb1-positive cells relative to overall cell numbers was observed in cultures seven days after glutamate treatment (unpaired two-tailed t-test, %Calb1Ctrl7d=43.1 ± 1.6, %Calb1Glu7d=36.7 ± 2.5, p=0.031; Fig. 4d-e).” (p.7, ll. 312-313)

  1. Discussion, page 16: “With a stable 92-96% neuronal phenotype, our cultures could be used for a nearly unbiased study of neuronal features of seizure generation”. Astrocytes heavily regulate neuronal synaptic communications. Furthermore, astrocytes are essential in the process of epileptogenesis. With >90% neuronal culture, will the neuronal physiology be reliably recapitulated?

Response: We agree with the reviewer regarding the importance of astrocytes for neuronal and network function, but we had no detectable evidence for their contribution. “Cell type attribution of detected signals was always confirmed by Map2/Gfap immunofluorescence of all calcium-imaged cultures. We identified specific but extremely rare events of glial calcium activity (i.e., very low frequency, high amplitude, slow propagating along glial processes), which only affected individual cells and never occurred in temporal relation with synchronized spiking or SLE development across the network.” (p.16, ll. 430-435)

Reviewer 2 Report

The topic of this manuscript is very interesting and probably worthwhile to study, but this work is still under preliminary stage.  It has missed many important analysis and discussion without which it should not be published in this journal. My specific comments are as follows:-

  1. In Fig 3: Dendritic sprouting patterns (the dendritic arborization pattern) should be measured using different parameters like branch number, shaft number etc. and compared between the control and experimental system. Statistical analysis should be shown.

Also, authors should include their own experimental evidence, and not just picture data from a previous published paper.

  1. To me Figure 4 data is also incomplete. Each and every experiment should have representative pictures for each time point (e.g. 3d, 7d) and statistical analysis (bar diagram with error bar and p value calculation from comparison using Student t test).
  2. Authors have included a whole set of WGBS data in the next figure. They also generate a list of genes that is important for brain development and changed methylation level under glutamate treatment. But 5 mc level change does not always result gene expression change. I suggest authors should chose some genes, preferentially related to Epilepsy, and check their expression level by IF and/or WB in glutamate treated neurons with proper control.
  3. Authors have just mentioned that their results confirm the previous reports of DNA methylation in epileptogenesis. I suggest authors should include data or at least discuss in more detail how their WGBS data can identify the similar or different sets of genes (with % similarity) with changed 5 mc level compared to previous data.
  4. Table 1 shows that glutamate treatment can change methylation level in many genes that are not known to be associated with Epilepsy. Rather, it shows change in methylation level in genes related to wide range of neurodegenerative and neurodevelopmental disorders. This, according to me, raise the question against the specificity of glutamate treatment in establishing ‘Epilepsy in dish’ and thus weaken the main claim of this manuscript. I think, this is a serious problem of this manuscript. Authors should give more evidence in support of their claim and discuss in detail how change in methylation level in genes related to other diseases lead to epilepsy like cellular phenomena e.g. SLE.
  5. In Introduction part authors should mention and discuss any previous attempts of establishing epilepsy disease cell model and why their work is important.

Author Response

Reviewer 2

The topic of this manuscript is very interesting and probably worthwhile to study, but this work is still under preliminary stage. It has missed many important analysis and discussion without which it should not be published in this journal. My specific comments are as follows:

  1. In Fig 3: Dendritic sprouting patterns (the dendritic arborization pattern) should be measured using different parameters like branch number, shaft number etc. and compared between the control and experimental system. Statistical analysis should be shown.

Response: Sholl analysis is an established and widely applied method to quantify structural changes of cells. We performed sholl analysis as described in the material and methods section (p.4, ll. 158-165) and quantified both dendritic branching and length. Statistics are provided in the text (p.8, ll.284-295), but a graphical summary is also shown in Fig. 2b.

Also, authors should include their own experimental evidence, and not just picture data from a previous published paper.
Response: Our own experimental evidence is summarized in Fig. 2. To avoid redundancies, we have now removed Fig. 3 from the manuscript and renumbered all subsequent Figures.

  1. To me Figure 4 data is also incomplete. Each and every experiment should have representative pictures for each time point (e.g., 3d, 7d) and statistical analysis (bar diagram with error bar and p value calculation from comparison using Student t test).

Response: Statistics including mean and SEM are reported in the main text (p.8, ll.300-301 and 312-313). We have also added box plots to NEW Fig. 3c, e.

  1. Authors have included a whole set of WGBS data in the next figure. They also generate a list of genes that is important for brain development and changed methylation level under glutamate treatment. But 5 mc level change does not always result gene expression change. I suggest authors should chose some genes, preferentially related to Epilepsy, and check their expression level by IF and/or WB in glutamate treated neurons with proper control.

Response: A temporal study on locus-specific (i.e., Gria2, Grin2a) multi-layered epigenetic gene regulation in this model was previously performed and published (Kiese et al. 2017). In the present study we focused on the broader question whether our model recapitulated a pattern of genome-wide DNA methylation changes as part of the epileptogenic process and development of seizure-like events. The present data support the idea that such genome-wide epigenetic changes are a common principle of the disease process irrespective of species, etiology, histopathology, or (NEW) model system. Such DNA methylation patterns may directly or indirectly account for some of the gene expression changes that contribute to the epileptic phenotype.

  1. Authors have just mentioned that their results confirm the previous reports of DNA methylation in epileptogenesis. I suggest authors should include data or at least discuss in more detail how their WGBS data can identify the similar or different sets of genes (with % similarity) with changed 5 mc level compared to previous data.

Response: A comprehensive integration of genomic DNA methylation data obtained in different rat epilepsy models (Methyl-Seq), human focal epilepsy (Methyl-Seq, Illumina arrays) and our present data (WGBS) has not yet been performed. Such integrated analysis to identify common epileptogenic mechanisms would be challenging with regards to species differences, annotation biases, platform variability, normalisation algorithms, cutoff points for selecting regulated genes, analysis of different brain structures and cells, use of variable insults to trigger epileptogenesis, selection of timepoints for sampling after the insult, and characterisation of epilepsy phenotype at the time of sampling (see also Response to 5.). We previously compared DNA methylation profiles obtained from Methyl-Seq in three different rat epilepsy models (i.e., same species, same time point/age, same profiling platform, same sequencing facility). Intriguingly, in the models studied, DNA methylation and gene expression profiles distinguished controls from epileptic animals, but there was no common methylation signature in all three different models and few regions common to any two models. Thus, the data provided evidence that genome-wide

alteration of DNA methylation signatures is a general pathomechanism associated with epileptogenesis and epilepsy in experimental animal models, but the broad pathophysiological differences between models (i.e. pilocarpine, amygdala stimulation, and post-TBI) were reflected in distinct etiology-dependent DNA methylation patterns (Debski et al., Sci Rep 2016).

  1. Table 1 shows that glutamate treatment can change methylation level in many genes that are not known to be associated with Epilepsy. Rather, it shows change in methylation level in genes related to wide range of neurodegenerative and neurodevelopmental disorders. This, according to me, raise the question against the specificity of glutamate treatment in establishing ‘Epilepsy in dish’ and thus weaken the main claim of this manuscript. I think, this is a serious problem of this manuscript. Authors should give more evidence in support of their claim and discuss in detail how change in methylation level in genes related to other diseases lead to epilepsy like cellular phenomena e.g. SLE.

Response: We thank the reviewer for this comment, which highlights a major challenge in epilepsy research. “Epilepsy is not a single disease, but highly heterogeneous with diverse clinical syndromes, associated etiologies (i.e., structural, genetic, infectious, metabolic, immune, unknown), and many different mechanisms contributing to the epileptic phenotype and known frequent comorbidities.”

Over 50% of people with epilepsy have one or more somatic (e.g., endocrine, inflammatory and autoimmune, metabolic, migraine and chronic pain, or sleep disorders) or psychiatric condition (e.g., autism, schizophrenia, depression). Bidirectional associations with epilepsy have been reported for several conditions including autism-spectrum disorders (neurodevelopmental) and Alzheimer’s disease (neurodegenerative; Lin et al., Lancet 2012). The complexity of epilepsy has further become evident from genetic studies. The number of epilepsy genes identified in monogenetic forms of epilepsies has grown rapidly and evolved from a handful of ion channel genes to now almost 150 pathogenic and likely pathogenic genes most of which are not involved in synaptic function and ion conductance of the cell, but in all aspects of metabolism, nuclear functions, development, or many other biological processes that cannot be easily linked to neuronal firing and seizures. Phenotypic pleiotropy is common which together with locus heterogeneity and allelic heterogeneity render genotype-phenotype correlations difficult (McTague et al., Lancet Neurol 2016). Likewise, in non-genetic focal epilepsies, gene expression studies in experimental animal models and human surgical tissue over decades failed to provide a single causative epilepsy gene or even biological pathway. Instead, hundreds of genes have been found deregulated and rarely there has been any gene found twice in repetitive studies (Lukasiuk & Pitkanen, Lancet Neurol 2011). Taken together, “the diversity of gene functions potentially affected by DNA methylation changes in our model are in line with the multifactorial nature of epilepsy and associated comorbidities and its link to brain development, function and ageing” (p.17, ll. 497-500).

Therefore, we have tried to be as unbiased as possible in the presentation of our molecular findings by showing the full range of gene functions affected by DNA methylation changes in our model.

  1. In Introduction part authors should mention and discuss any previous attempts of establishing epilepsy disease cell model and why their work is important.

Response: We highly appreciate the comment. While the introduction to our manuscript cannot provide a complete and comprehensive discussion of any previous attempts to model epilepsy, it has now been extended follows: “While seizures are commonly defined as the clinical manifestation of an abnormal, excessive, hypersynchronous discharge of a population of cortical neurons [16], it remains unclear to date what the minimum cellular and molecular requirements are that induce and promote the complex sequence of events leading to seizure development. Many experimental animal models have been developed in the past to induce and study acute seizures or epileptogenesis [35], e.g., following neonatal hyperthermia [14, 23], hypoxia [3, 10, 26, 27, 47], traumatic brain injury [12, 20, 38], tetanus toxin injection [1, 25, 44], pilo-carpine induced SE [17, 42], kainic acid induced SE [53], or electrical kindling [41]. However, animal models have a limited ability to recapitulate human pathologies even though they appear to symptomatically mimic human disease. They are further are highly cost, labor, and time consuming. Also the complexity of the mammalian brain limits the possibilities to address questions of cause and consequence and it remains challenging to attribute specific disease pathomechanisms over time to individual cell populations. In vitro models, including acute hippocampal slice preparations, organotypic hippocampal slice cultures [39], and iPSC-based brain organoid models [12] may be used to address some of these limitations. Moreover, dissociated primary neuronal cultures and neuronal cell lines are used to study basic electrophysiological properties and screen, e.g., for neurotoxicity of certain compounds [8, 18, 46].” (p.2, ll. 62-79)

Round 2

Reviewer 1 Report

I thank the authors for all the clarification and I believe this manuscript is ready for publication.

Author Response

We kindly thank this reviewer for their positive feedback!

Reviewer 2 Report

This reviewer sadly observed that authors are reluctant to revise their current manuscript according to the suggestions. In this case I shall request the editor to take the decision about this manuscript. My specific replies on authors comments are in red and as as follows:

  1. In Fig 3: Dendritic sprouting patterns (the dendritic arborization pattern) should be measured using different parameters like branch number, shaft number etc. and compared between the control and experimental system. Statistical analysis should be shown.

Response: Sholl analysis is an established and widely applied method to quantify structural changes of cells. We performed sholl analysis as described in the material and methods section (p.4, ll. 158-165) and quantified both dendritic branching and length. Statistics are provided in the text (p.8, ll.284-295), but a graphical summary is also shown in Fig. 2b.

Using Sholl analysis or any other method hippocampal dendritic arborizing pattern of normal and patient or animal model with disease phenotypes should be measured and compared with the cell model system of schizophrenia (glutamate treated cells) described in this report.

Also, authors should include their own experimental evidence, and not just picture data from a previous published paper.
Response: Our own experimental evidence is summarized in Fig. 2. To avoid redundancies, we have now removed Fig. 3 from the manuscript and renumbered all subsequent Figures.

In the previous form of manuscript authors wrote………Structural comparison of neurons from our in vitro epileptogenic networks with previously published preparations in human 3D reconstructions of confocal laser scanning microscopy images of Lucifer Yellow dye-injected neurons in human epileptic hippocampal tissue [5] identified morphological similarities of the dendritic arborization pattern (Fig. 3).

In the revised version they omitted Figure 3. Since authors are trying to establish a new cell model system of schizophrenia, I believe, comparison of authors data (control vs glu treatment) with human epileptic tissue or tissue from previously established animal model system is necessary. And the dendritic arborization analysis from patients or from animal model should be done by authors and should be un-published.                                    

  1. To me Figure 4 data is also incomplete. Each and every experiment should have representative pictures for each time point (e.g., 3d, 7d) and statistical analysis (bar diagram with error bar and p value calculation from comparison using Student t test).

Response: Statistics including mean and SEM are reported in the main text (p.8, ll.300-301 and 312-313). We have also added box plots to NEW Fig. 3c, e.

OK

  1. Authors have included a whole set of WGBS data in the next figure. They also generate a list of genes that is important for brain development and changed methylation level under glutamate treatment. But 5 mc level change does not always result gene expression change. I suggest authors should chose some genes, preferentially related to Epilepsy, and check their expression level by IF and/or WB in glutamate treated neurons with proper control.

Response: A temporal study on locus-specific (i.e., Gria2, Grin2a) multi-layered epigenetic gene regulation in this model was previously performed and published (Kiese et al. 2017). In the present study we focused on the broader question whether our model recapitulated a pattern of genome-wide DNA methylation changes as part of the epileptogenic process and development of seizure-like events. The present data support the idea that such genome-wide epigenetic changes are a common principle of the disease process irrespective of species, etiology, histopathology, or (NEW) model system. Such DNA methylation patterns may directly or indirectly account for some of the gene expression changes that contribute to the epileptic phenotype.

This reviewer thinks to link a high throughput data analysis with some biological events e.g. seizure-like events during schizophrenia, further evidence is necessary. So, I still suggest to check part of these data by molecular experiments e.g. WB and/or analysis.

  1. Authors have just mentioned that their results confirm the previous reports of DNA methylation in epileptogenesis. I suggest authors should include data or at least discuss in more detail how their WGBS data can identify the similar or different sets of genes (with % similarity) with changed 5 mc level compared to previous data.

Response: A comprehensive integration of genomic DNA methylation data obtained in different rat epilepsy models (Methyl-Seq), human focal epilepsy (Methyl-Seq, Illumina arrays) and our present data (WGBS) has not yet been performed. Such integrated analysis to identify common epileptogenic mechanisms would be challenging with regards to species differences, annotation biases, platform variability, normalisation algorithms, cutoff points for selecting regulated genes, analysis of different brain structures and cells, use of variable insults to trigger epileptogenesis, selection of timepoints for sampling after the insult, and characterisation of epilepsy phenotype at the time of sampling (see also Response to 5.). We previously compared DNA methylation profiles obtained from Methyl-Seq in three different rat epilepsy models (i.e., same species, same time point/age, same profiling platform, same sequencing facility). Intriguingly, in the models studied, DNA methylation and gene expression profiles distinguished controls from epileptic animals, but there was no common methylation signature in all three different models and few regions common to any two models. Thus, the data provided evidence that genome-wide

alteration of DNA methylation signatures is a general pathomechanism associated with epileptogenesis and epilepsy in experimental animal models, but the broad pathophysiological differences between models (i.e. pilocarpine, amygdala stimulation, and post-TBI) were reflected in distinct etiology-dependent DNA methylation patterns (Debski et al., Sci Rep 2016).

  1. Authors should discuss it in Discussion and Result section in detail.
  1. Table 1 shows that glutamate treatment can change methylation level in many genes that are not known to be associated with Epilepsy. Rather, it shows change in methylation level in genes related to wide range of neurodegenerative and neurodevelopmental disorders. This, according to me, raise the question against the specificity of glutamate treatment in establishing ‘Epilepsy in dish’ and thus weaken the main claim of this manuscript. I think, this is a serious problem of this manuscript. Authors should give more evidence in support of their claim and discuss in detail how change in methylation level in genes related to other diseases lead to epilepsy like cellular phenomena e.g. SLE.

Response: We thank the reviewer for this comment, which highlights a major challenge in epilepsy research. “Epilepsy is not a single disease, but highly heterogeneous with diverse clinical syndromes, associated etiologies (i.e., structural, genetic, infectious, metabolic, immune, unknown), and many different mechanisms contributing to the epileptic phenotype and known frequent comorbidities.”

Over 50% of people with epilepsy have one or more somatic (e.g., endocrine, inflammatory and autoimmune, metabolic, migraine and chronic pain, or sleep disorders) or psychiatric condition (e.g., autism, schizophrenia, depression). Bidirectional associations with epilepsy have been reported for several conditions including autism-spectrum disorders (neurodevelopmental) and Alzheimer’s disease (neurodegenerative; Lin et al., Lancet 2012). The complexity of epilepsy has further become evident from genetic studies. The number of epilepsy genes identified in monogenetic forms of epilepsies has grown rapidly and evolved from a handful of ion channel genes to now almost 150 pathogenic and likely pathogenic genes most of which are not involved in synaptic function and ion conductance of the cell, but in all aspects of metabolism, nuclear functions, development, or many other biological processes that cannot be easily linked to neuronal firing and seizures. Phenotypic pleiotropy is common which together with locus heterogeneity and allelic heterogeneity render genotype-phenotype correlations difficult (McTague et al., Lancet Neurol 2016). Likewise, in non-genetic focal epilepsies, gene expression studies in experimental animal models and human surgical tissue over decades failed to provide a single causative epilepsy gene or even biological pathway. Instead, hundreds of genes have been found deregulated and rarely there has been any gene found twice in repetitive studies (Lukasiuk & Pitkanen, Lancet Neurol 2011). Taken together, “the diversity of gene functions potentially affected by DNA methylation changes in our model are in line with the multifactorial nature of epilepsy and associated comorbidities and its link to brain development, function and ageing” (p.17, ll. 497-500).

Therefore, we have tried to be as unbiased as possible in the presentation of our molecular findings by showing the full range of gene functions affected by DNA methylation changes in our model.

 OK

  1. In Introduction part authors should mention and discuss any previous attempts of establishing epilepsy disease cell model and why their work is important.

Response: We highly appreciate the comment. While the introduction to our manuscript cannot provide a complete and comprehensive discussion of any previous attempts to model epilepsy, it has now been extended follows: “While seizures are commonly defined as the clinical manifestation of an abnormal, excessive, hypersynchronous discharge of a population of cortical neurons [16], it remains unclear to date what the minimum cellular and molecular requirements are that induce and promote the complex sequence of events leading to seizure development. Many experimental animal models have been developed in the past to induce and study acute seizures or epileptogenesis [35], e.g., following neonatal hyperthermia [14, 23], hypoxia [3, 10, 26, 27, 47], traumatic brain injury [12, 20, 38], tetanus toxin injection [1, 25, 44], pilo-carpine induced SE [17, 42], kainic acid induced SE [53], or electrical kindling [41]. However, animal models have a limited ability to recapitulate human pathologies even though they appear to symptomatically mimic human disease. They are further are highly cost, labor, and time consuming. Also the complexity of the mammalian brain limits the possibilities to address questions of cause and consequence and it remains challenging to attribute specific disease pathomechanisms over time to individual cell populations. In vitro models, including acute hippocampal slice preparations, organotypic hippocampal slice cultures [39], and iPSC-based brain organoid models [12] may be used to address some of these limitations. Moreover, dissociated primary neuronal cultures and neuronal cell lines are used to study basic electrophysiological properties and screen, e.g., for neurotoxicity of certain compounds [8, 18, 46].” (p.2, ll. 62-79)

OK
